# Improving Visual Function after Mild Traumatic Brain Injury Using a Vision Therapy Program: Case Reports

**DOI:** 10.3390/brainsci10120947

**Published:** 2020-12-07

**Authors:** Mona-Lisa Möller, Susanna Melkas, Jan Johansson

**Affiliations:** 1Neurology, University of Helsinki and Helsinki University Hospital, 00014 Helsinki, Finland; susanna.melkas@hus.fi; 2Department of Clinical Neuroscience, Eye and Vision, Karolinska Institutet, 17177 Stockholm, Sweden; jan.johansson.1@ki.se

**Keywords:** mild traumatic brain injury (MTBI), vision therapy, CISS

## Abstract

This case report describes the outcome of vision therapy for three patients who were referred to therapy due to visual symptoms after mild traumatic brain injury (MTBI). The criterion for inclusion was a high score (>21p) on the Convergence Insufficiency Symptom Survey (CISS) scale. The vision therapy program (VTP) included both face-to-face sessions and home-based tasks. Cases #1 and #2 had a substantial CISS scale evaluation improvement, and case #2 normalized the CISS scale score from 36 to 19. All patients agreed that vision therapy helped them understand their own vision and changes in their vision, which helped their overall recovery after MTBI. Rehabilitation professionals have an important role in screening for vision impairments and treating functional vision challenges after mild traumatic brain injury.

## 1. Introduction

Visual disturbances due to traumatic brain injury are common, with incidence varying from 30% to 85% [1]. They include anomalies of accommodation, version and vergence, eye movements, photosensitivity and ocular health issues [1,2]. Accommodative dysfunction is referred to as insufficiency and infacility, a reduced ability to achieve, maintain or alter focus to see clearly. Version deficits include abnormal fixation, pursuits and saccades, i.e., impaired or less efficient conjugated eye movements, for example, those used when reading. Vergence dysfunction includes convergence insufficiency, reduced fusional vergence or an affected ability to maintain alignment of the eyes for single clear vision [1].

Accommodative dysfunctions can lead to intermittent blurs in vision-related tasks such as reading. Version deficits may influence reading-related tasks that manifest as losing one’s place while reading, skipping or re-reading words and reduced, inefficient reading. Vergence deficits may lead to double vision, eyestrain or brow ache, motion sensitivity and shimmering vision [3].

A large review of vision rehabilitation interventions following mild traumatic brain injury by Simpson-Jones and Hunt, published in 2019 [4], showed that promising interventions are emerging in this field but more research is needed on the different methods. In the review, the research interventions for vision rehabilitation, following mild traumatic brain injury, were divided into three categories: optical device therapy, vision or oculomotor therapy and a combination of optical devices and vision therapy.

The aim of our study was to obtain an understanding for the possibility to improve visual function after mild traumatic brain injury (MTBI) by vision therapy.

## 2. Materials and Methods

This case report describes the outcome of vision therapy for three patients who were referred to therapy due to visual symptoms after MTBI. The criterion for inclusion was a high score (>21) on the Convergence Insufficiency Symptom Survey (CISS) scale. CISS was originally intended to evaluate vision symptoms in non-brain-injured pediatric populations [5]. It has since been evaluated in an adult population [6] in which a total score of 21 or higher indicates an elevated level of visual symptoms. The patients were followed up by a neurologist at the Traumatic Brain Injury Outpatient Clinic, Helsinki University Hospital (HUS). The results are presented by outcomes from clinical measures of visual function and scores from the CISS scale. The cases are part of a larger cohort of patients recruited for a study on vision therapy, with a one-year follow-up and with more detailed outcomes. All included patients gave their written consent. This study was approved by the Ethics Committee of Helsinki University Hospital (dnro 16.08.2017 §154 and 27.11.2019 §192).

We used the World Health Organization (WHO) definition of MTBI [7]. These criteria include (1) one or more of the following: confusion or disorientation, loss of consciousness for 30 min or less, post-traumatic amnesia for less than 24 h and/or other transient neurological abnormalities such as focal signs, seizure and intracranial lesion not requiring surgery; and (2) a Glasgow Coma Scale score of 13–15 after 30 min post-injury or later upon presentation for healthcare. These manifestations of MTBI must not be due to drugs, alcohol or medications, or caused by other injuries or treatments for other injuries (e.g., systemic injuries, facial injuries or intubation), other problems (e.g., psychological trauma, language barrier or coexisting medical conditions) or penetrating craniocerebral injury.

Upon inclusion in the study, an examination of visual function was performed, which included visual acuity, near point of convergence, convergence facility, near point of accommodation, accommodative facility and fusional vergence width at near (Table 1). The examination was repeated after the vision therapy program. The criteria for normal function were set by normative standards derived from the literature [8,9,10]. To test the basic visual acuity, the Lea number line test was used and interpreted by the normative values according to international recommendations. Near point of accommodation was interpreted by the normative values in centimeters according to age.

The vision therapy program (VTP) included both face-to-face sessions and home-based tasks. The patients were targeted with customized, specific vision therapy activities, based on the findings in the vision examination, with assistance from the vision therapist. Each training session consisted of the following areas according to the method developed by Wilhelmsen [11]:Warm-up tasks to make the patient aware of their vision problems and to help them understand the importance of using eye movements.Vision search over a broad area to work on the eye scanning, tracking and searching across farther distances.Fast eye movements in all directions to improve saccadic capacity.Tasks intended to improve capacity for near activities, such as reading. The different tasks aimed to improve accommodation, vergence functions, binocularity, saccades, visual search, tracking and reading speed.

All tasks were performed both monocularly and binocularly except the ones that needed both eyes such as binocularity and fusion.

Warm-up tasks and visual search training included tasks finding objects with certain criteria (letter, number and word cards, memory cards) both monocularly and binocularly at varying distances. Fast eye movements were trained using rapid and precise eye movements and fixation, sometimes using a metronome to control the pace of the movements from target to target. The tasks to improve the capacity for near vision activities such as reading were trained as follows: vergence-related deficits were directed with free space training (Brock´s string, Hart chart) and computerized training (VISIOcoach and Vision Builder); and fixation, accommodation and saccades were directed with systematic exercises requiring the patient to make precise eye movements and quickly achieve a steady fixation both monocularly and binocularly (Hart chart, Saccadic strips, Flipper lenses, find-the-differences cards, etc.). All the visual tasks emphasized no head movements, therefore allowing only eye movements. Every training activity during the session lasted for 10 min, with a short break before changing to the next activity. During the session, activities varied between seated or standing tasks.

## 3. Our Patients

### 3.1. Case 1

A 30-year-old male patient was referred for visual function assessment 16 months after trauma. The diagnosis was MTBI and findings in MRI were two single microhemorrhages. He scored 47 on the CISS scale (symptomatic > 21). The patient noted the following vision problems that he experienced daily: nausea when moving, difficulty reading and light sensitivity. With the first case, 15 one-hour face-to-face sessions were evenly distributed over a three-week period. Table 2 shows the results before and after the three-week intense face-to-face therapy results. All the measured near task oculomotor functions improved except the near point of convergence, which was good before the rehabilitation.

The patient stated the following regarding his own experience in vision therapy: “When we started vision therapy, it was assessed that my eyes were at the level of a 75-year-old person. They simply didn’t work together and felt strange. Especially the left eye that probably was shaken up a lot because of my tackle… In vision therapy I got answers and could put in words what I felt but had not been able to express myself”.

### 3.2. Case 2

A 40-year-old female patient was referred for visual function assessment with visual therapy 12 months after trauma. Her diagnosis was MTBI without any findings in the MRI, but with a sprain of ligaments in the cervical spine. She had a CISS score of 36 (symptomatic > 21). The patient referred to problems reading texts, memorizing the reading and problems with right eye visual function, potentially visual agnosia. Vision therapy with the second case was 10 one-hour face-to-face sessions and 5 home training sessions over a three-week period. Table 3 shows the results before and after the three-week intense vision therapy results. All the measured near task oculomotor functions normalized or improved.

A huge improvement can be seen especially in the patient´s own experience measured by the CISS scale. She comments, “When I started working with vision therapy, my situation was complicated. After a small car accident, my vision changed, and with it, a large part of my life, because my visual memory represents about 80% of my work and my daily memories. My eyes were unfocused, and sometimes, I did not recognize what appeared on the right side; my cat, for example, if she came stealthily from that side, I didn´t recognize that it was her because I saw a blur that appeared suddenly, and sometimes it scared me. Driving was also very unsafe, because sometimes, visual inputs appeared coming from the right that I couldn´t recognize. The sessions and exercises with vision therapy were excellent! It helped me to understand how my eyes worked and what had changed, and thanks to this, I did my recovery exercises properly. Although it had been a long time since my accident, we took a super strong step together.”

### 3.3. Case 3

A 55-year-old female patient was referred to the visual therapy program and the visual function assessment 12 months after trauma. The diagnosis was MTBI without any findings in the MRI, but with a sprain of ligaments in the cervical spine. The CISS score was 42 (symptomatic > 21). According to the patient, the main problems due to vision were light sensitivity, reading problems and perceiving information on the right visual field. Vision therapy with the third case included 2 one-hour face-to-face sessions and 20 home training sessions over a period of six months. Table 4 shows the results before and after vision therapy. Near distance fusion width and near point of convergence normalized. Near point and rotation accommodation improved.

The CISS scores remained the same as before the vision therapy but improvements could be seen in all the near visual functions measured. The patient commented, *“My visual problem after the TBI was difficulty finding things on the computer screen, especially when scrolling the text/screen. Lines were mixed and became shadowed. After the visual therapy symptoms were slightly alleviated but did not disappear totally. I would have preferred several appointments so that there would have been more time to implement new tasks and methods”*.

## 4. Discussion

The results indicate that vision therapy may strengthen oculomotor function and can make a difference in near task visual functioning. Although it seems that face-to-face sessions are not obligatory, they were associated with more pronounced improvement, together with an intense three-week rehabilitation (cases #1 and #2). Cases #1 and #2 had a substantial CISS scale evaluation improvement, and case #2 normalized the CISS scale score from 36 to 19. The implementation of the visual therapy took place about 1 year after the brain injury for cases #1 and #2 and more than two years after the trauma for case #3. Despite the fact that the time since trauma was more than one year (case #2) and 16 months (case #1), it shows that it is possible to influence and improve visual function. Even in case #3, with only 2 one-hour face-to-face sessions and 20 home training sessions over a period of six months, there was also improved fusion, convergence and accommodation. It has been suggested that restorative methods through intense training can re-activate residual neurons within or at the borders of a damaged area caused by a traumatic brain injury [12]. In this study, the visual changes were seen as clinically significant improvements. All the patients also mentioned that vision therapy helped them understand their own vision and changes in their vision after the trauma.

## 5. Summary

In this study, we used vision or oculomotor therapy as a rehabilitation intervention following mild traumatic brain injury. The comments from the patients in this study are congruent with the conclusion of the review by Simpson-Jones and Hunt [4]: rehabilitation professionals have an important role in screening for vision impairments and treating functional vision challenges after mild traumatic brain injury.

## Figures and Tables

**Table 1 brainsci-10-00947-t001:** Visual function and method of examination with criteria for normal function.

Visual Function	Method of Examination	Criteria for Normal Function
Visual acuity	Visual chart Lea numbers at 40 cm	≥0.8 normal with best correction
Near point of convergence	Push-up method using the convergence fixation target on a Royal Air Force Near Point Rule (RNPR)	≤6 cm [10]
Convergence facility	3 base in (BI) 12 base out (BO) prism	Age < 40 years: ≥13.5 cycles per minute (CPM) [8] Age ≥ 40 years: ≥7 CPM [9]
Near point of accommodation	Push-up method using the near chart target on an RNPR	Age-related mean score; 30 y ≤ 11.5 cm and 40 y ≤ 17.5 cm
Accommodative facility	Spherical flipper ±1.0 or ±1.5 diopters	>10 cycles per minute [10]
Fusional vergence width at near viewing	Prism bar	≤27 prism diopters (PD) [10]

**Table 2 brainsci-10-00947-t002:** Total CISS score and clinical measures for patient #1. Only measures with subnormal values before vision therapy are presented.

	Before VTP	After VTP	Comment
CISS score	47	34	improvement
Clinical measures;			
Near distance fusion width, cm	6	12	improvement
Near point of convergence, cm	8	8	-
Convergence facility 3BI/12BO, CPM	0	2	improvement
Near point of accommodation bin, cm	16	12	normalization
Accommodation facility 1.5D, CPM	10	18	normalization

CISS = convergence insufficiency symptom survey, BI = base in, BO = base out, CPM = cycles per minute, VTP = vision therapy program.

**Table 3 brainsci-10-00947-t003:** Total CISS score and clinical measures for patient #2. Only measures with subnormal values before vision therapy are presented.

	Before VTP	After VTP	Comment
CISS score	36	19	normalized
Clinical measures;			
Near distance fusion width, cm	12	41	normalized
Near point of convergence, cm	10	6	normalized
Convergence rotation 3BI/12BO, CPM	0	7	normalized
Near point of accommodation bin, cm	30	26	improvement
Accommodation rotation 1.0D, CPM	0	10	improvement

CISS = convergence insufficiency symptom survey, BI = base in, BO = base out, CPM = cycles per minute, VTP = vision therapy program.

**Table 4 brainsci-10-00947-t004:** Total CISS score and clinical-based measures for patient #3. Only measures with subnormal values before the vision therapy program are presented.

	Before VTP	After VTP	Comment
CISS score	42	42	-
Clinical measures;			
Near distance fusion width, cm	26	32	normalized
Near point of convergence, cm	12	6	normalized
Convergence rotation 3BI/12BO, CPM	4	9	normalized
Near point of accommodation bin, cm	25	18	improvement
Accommodation rotation, CPM	-	-	not measurable due to age

CISS = convergence insufficiency symptom survey, BI = base in, BO = base out, CPM = cycles per minute, VTP = vision therapy program.

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
