# Peer review of "Improving Visual Function after Mild Traumatic Brain Injury Using a Vision Therapy Program: Case Reports"

_brainsci, 2020, doi:10.3390/brainsci10120947_

Round 1

Reviewer 1 Report

I would like to commend the authors on a very well written and clearly presented study; I admire the concise and coherent phrasing, and methods and results which are direct and easy to understand. This is an interesting case series relating to the rehabilitation practices for individuals with various visual/ocular symptoms following a traumatic brain injury and does add to the literature. There are a few aspects I would like to see addressed in order for the paper to be improved: 

1) The background information provides excellent detail regarding the nature of visual symptoms, but the nature of the study/study aims are not clear from this. In the discussion there is reference to previous literature and a need for more research in the area of visual rehabilitation after TBI, and I think that more of this needs to be discussed earlier on to better set the scene for the study. 

2) It is mentioned that this is a case series from a larger cohort study. I would like more details on what exactly is being explored here that isn't covered there (e.g., are the outcomes more detailed? - I assume so but would like this to be explicit). 

3) Page 2, line 72: there is a typo ('nearvision' should be 'near vision'). 

4) Lines 144 - 146. "This suggests that restorative methods through intense training can re-activate residual neurons within or at the borders of a damaged area". I think that this conclusion is a little misleading based on the evidence collected; you have functional evidence of improvement, but no data (e.g,. neuroimaging) to demonstrate the neurological mechanisms that may underlie this. 

Author Response

Response to Reviewers
Reviewer 1:
Please see attachment.

Reviewer 2 Report

This is an interesting 3-case clinical presentation of the use of visual training in the resolution of visual symptomatology and dysfunctions associated with Traumatic Brain Injury (TBI). Some effectiveness due to VT is demonstrated including the patient educational value regarding each patient's personal syndrome and how such education is valuable in patient understanding and thus tolerance of their condition beyond any clinical improvements that may have been seen. This paper has value as a clinical description of what VT after TBI looks like and how it may be judged to have been successful. It should not be the goal of these authors to "prove" anything which such a small sample of clinical subjects with such a variable outcome for each. Nevertheless, the basic clinical improvements in each case signal the underlying value of working with such cases even after long periods post-injury. Again the importance of the gestalt of the training circumstance is evident here and is valuable to report as are the physical improvements that were seen. Please see comment bubbles throughout the text for ideas to make specific textual improvements that will aid the reader's understanding.

Author Response

Reviewer 2, 
